

# Harvesting and chewing as constraints to forage consumption by the African savanna elephant (*Loxodonta africana*)

Bruce W. Clegg[1,2] and Timothy G. O'Connor[1,3]

[1] School of Animal, Plant and Environmental Sciences, University of the Witwatersrand, Johannesburg, South Africa
[2] The Malilangwe Trust, Chiredzi, Zimbabwe
[3] South African Environmental Observation Network (SAEON), Pretoria, South Africa

## ABSTRACT

As a foundation for understanding the diet of African savanna elephants (*Loxodonta africana*), adult bulls and cows were observed over an annual cycle to determine whether harvesting ($P_t$), chewing ($C_t$) and handling times ($H_t$) differed across food types and harvesting methods (handling time is defined as the time to harvest, chew and swallow a trunkload of food). Bulls and cows were observed 105 and 26 times, respectively (94 and 26 individuals), with a total of 64 h of feeding recorded across 32 vegetation types. Some food types took longer to harvest and chew than others, which may influence intake rate and affect choice of diet. The method used to gather a trunkload of food had a significant effect on harvesting time, with simple foraging actions being comparatively rapid and more difficult tasks taking longer. Handling time was constrained by chewing for bulls, except for the processing of roots from woody plants, which was limited by harvesting. Time to gather a trunkload had a greater influence on handling time for cows compared to bulls. Harvesting and handling times were longer for bulls than cows, with the sexes adopting foraging behaviors that best suited their energy requirements.

## INTRODUCTION

African savanna elephants (*Loxodonta africana*) utilize a wide variety of forage types, consuming leaves, stems, roots and tubers from herbaceous vegetation (grass and forbs) (*Barnes, 1982*; *De Boer et al., 2000*; *de Longh et al., 2004*; *Wyatt & Eltringham, 1974*), and leaves, twigs, bark, roots, flowers and fruits from woody plants (*Field, 1971*; *Guy, 1976*; *Owen-Smith & Chafota, 2012*). Although elephants harvest food from a range of plant life forms, it is their conspicuous impact on woodlands that has the greatest potential to cause long-term vegetation change (*Lamprey et al., 1967*; *Laws, 1970*; *Leuthold, 1977*; *Morrison, Holdo & Anderson, 2016*). Extensive conversion of woodlands to shrubland by elephants (*Spinage, 1994*) and the potential associated loss of biodiversity (*Cumming et al., 1997*; *Herremans, 1995*; *Kerley & Landman, 2006*) may require management intervention (*O'Connor, Goodman & Clegg, 2007*), but any action should be based on an

Corresponding author
Bruce W. Clegg,
bruce@malilangwe.org

understanding of why elephants choose to utilize woody plants in a destructive manner. Impact on woody vegetation is greatest when harvesting methods such as breaking branches, debarking stems, or toppling, pollarding or uprooting whole plants are used, and less when trunkloads of leaves are stripped without breaking branches (*O'Connor, Goodman & Clegg, 2007*). When diet is composed solely of grass and forbs there is no damage to woody plants. Therefore knowledge of the factors that influence choice of diet and mode of harvesting is required before impact on woody vegetation can be understood.

Elephants spend up to 18 h per day foraging (*Wyatt & Eltringham, 1974*). This involves locating suitable food patches, harvesting trunkloads, and chewing and swallowing harvested material. The time to complete these individual tasks may vary across forage types, potentially causing differences in the rate at which each forage type can be ingested. This in turn may influence diet and habitat selection because elephants possibly seek to maximize their rate of intake of food rich in easily digestible cell solubles (*Clegg, 2010*; *O'Connor, Goodman & Clegg, 2007*). Food types that can be located, harvested and chewed quickly should have higher preference than those that take longer to ingest. Determinants of searching time (time to locate a food patch) have been investigated for elephants (B. Clegg, 2016, unpublished data), but the potential for differences in harvesting and chewing times across the range of forage types consumed is yet to be explored. Adult females have half the body mass of adult males (*Owen-Smith, 1988*) and this may cause differences in the strength and capacity available for harvesting and chewing fibrous food types. Consequently, the possibility that gender influences time to harvest and chew food is also explored in this study.

Early foraging models assumed that harvesting and chewing by herbivores were mutually exclusive processes (*Farnsworth & Illius, 1996*; *Farnsworth & Illius, 1998*; *Spalinger & Hobbs, 1992*). However, harvesting and chewing have been shown to overlap for both cattle (*Laca, Ungar & Demment, 1994*) and giraffe (*Ginnett & Demment, 1995*). This is also true for elephants because the trunk allows harvesting to take place while food is being chewed (*Clegg, 2010*). This overlap means that handling time (time to harvest, chew and swallow a trunkload of food; $H_t$) is constrained by either harvesting time ($P_t$) or chewing time ($C_t$) depending on which action takes longest (i.e. when $P_t > C_t$, $H_t = P_t$, but when $H_t > P_t$, $H_t = C_t$) (*Clegg, 2010*). Therefore harvesting methods that involve laborious, time-consuming actions may considerably lower the rate of food intake even if trunkloads are rapidly chewed and swallowed. This study focused on foraging within a food patch. Movement between patches necessitates the inclusion of searching time as an additional constraint, and this is dealt with elsewhere using a more complete foraging model (*Clegg, 2010*).

The aim of this study was to determine whether harvesting, chewing and handling time differ across food types as a foundation for understanding diet choice of elephants. The following specific questions were addressed: (1) Do some forage types take longer to harvest and chew than others; (2) Does the method used to gather a trunkload affect harvesting time; (3) Is handling time constrained by harvesting or chewing; (4) Does gender influence handling time?

## MATERIALS AND METHODS

### Study area

The study was conducted in the semi-arid savanna of Malilangwe Wildlife Reserve (50,000 ha) in south-eastern Zimbabwe (20°58′–21°15′S, 31°47′–32°01′E). Permission to carry out the research was granted by The Malilangwe Trust. Ethical approval was not required because elephants were simply observed in the wild and not interfered with in any way. The reserve has a hot wet season from November to March, a cool dry season from March to August, and a hot dry season from September to October. Mean annual rainfall is 557 mm ($n = 64$; $CV = 34.2\%$), with approximately 84% falling in the hot wet season. Rainfall during the year of study was 716 mm. The average minimum and maximum monthly temperatures range from 13.4 °C (July) to 23.7 °C (December), and 23.2 °C (June) to 33.9 °C (November), respectively (*Clegg, 2010*). Frost is rare. Thirty-eight vegetation types, from open grassland to dry deciduous forest, have been identified on seven geological types, with soils ranging from 90% sand to 41% clay (*Clegg & O'Connor, 2012*).

### Data collection and analysis

Harvesting time ($P_t$) and handling time ($H_t$) were estimated for different food types by observing elephants feeding between April 2002 and March 2003. Chewing time ($C_t$) was not estimated directly, but because of the potential for complete overlap between harvesting and chewing, when $H_t > P_t$, it was assumed that $H_t = C_t$. Observations were made in as many vegetation types and times of day as possible. No observations were made at night. Once elephants were located, a focal individual was chosen. Random selection was impossible because of the dictates of wind direction, availability of cover for an undetected approach and the presence of other elephants, and therefore selection was restricted to the most accessible adult (approximately > 30 years old). The sex of the focal animal was recorded and characteristics such as tusk length, shape and size, and torn ears were noted to ensure recognition during sampling. Observations were made on foot or from a vehicle for the larger family groups at a distance of 20–50 m using binoculars. The time at the start of the feeding record was noted. The following was recorded for each trunkload by talking at the instant of each foraging action into a head-set microphone attached to a dictaphone that was running continuously: (1) when the elephant began to harvest a trunkload; (2) harvesting method; (3) forage type; (4) plant species; and (5) when the trunkload was placed in the mouth. The point at which the elephant finished chewing a mouthful was assumed to take place the instant before the next trunkload was placed in the mouth. It was also noted when the elephant left a patch of food and started to feed in a new patch. The elephant was deemed to have left a woody patch if it abandoned the shrub or tree it had been feeding on or an herbaceous patch if it walked more than two paces without feeding from the herbaceous layer. If the focal elephant disappeared from view, recording was stopped. Recording continued when the elephant reappeared. If it became obvious that the elephant was walking to water as opposed to actively feeding, or if feeding was disturbed in any other way, the observation

was abandoned. The route and distance travelled during the observation period was recorded by saving a track on a Global Positioning System. The dictaphone recordings were transferred to a computer where they were analyzed using Winamp® (a digital audio player) and Microsoft Excel. Because the dictaphone was running continuously, the recording preserved the intervals between feeding actions. Consequently when recordings were played using Winamp® the time at the start and end of each feeding action could be read to the nearest second off the digital timer. These times were transferred to Excel spreadsheets that were used to construct data sets for $P_t$ and $H_t$ that included forage type, harvesting method, vegetation type, month, and elephant gender and ID for each trunkload. $H_t$ was calculated as the interval between consecutive mouthfuls of a food type gathered from a single patch using the same method of harvesting.

Many combinations of forage type and harvesting method had insufficient observations for analysis and therefore a single categorical variable called "Ftype" that included the nine most common combinations (pluck green grass, pluck and shake mixed grass, pluck green forbs, strip green leaves from woody plants, pluck leaves and twigs from woody plants, chew off bark from canopy branches, remove bark from the main stem of trees, break off root from woody plants, dig and break off root from woody plants) was constructed. To account for spatial, temporal, and within subject non-independence of observations we used the glmer function of the lme4 package of R (*Bates et al., 2015*) to create generalized linear mixed-effects models (GLMM) with harvesting or handling time as the dependent variable, and forage type (factor with nine levels) and sex (factor with two levels) as fixed effects. Vegetation type (spatial non-independence), month (temporal non-independence), and elephant ID (within subject non-independence) were used as crossed, uncorrelated, random intercept effects. Models failed to converge when slope was included in the structure of random effects. Therefore, only random intercepts were considered. Distributions of harvesting and handling times were right skewed so models were specified with the gamma distribution and log link to achieve homoscedasticity of residuals. The interaction between main effects was not included because data were missing for some forage type and gender combinations. Models with all possible groupings of random intercept effects were compared by assessing goodness of fit using Akaike (AIC) and Bayesian (BIC) information criteria acquired using the *AIC* (*R Core Team, 2016*) and *BIC* (*Pinheiro et al., 2016*) functions of R respectively. The *Anova* (*Fox & Weisberg, 2011*) and *anova* (*R Core Team, 2016*) functions, and *lsmeans* package (*Lenth, 2016*) of R were then run on the outputs of the best models to determine the significance of the fixed effects and calculate the least squares means of harvesting and handling time (and 95% confidence intervals) for the different forage type and gender combinations. The *lsmeans* package was used to conduct pairwise comparisons of the least squares means across forage types using Tukey's adjustment. Within forage types, we tested for a significant difference between harvesting and handling times by calculating the 95% confidence interval of the difference (intervals that included zero were not significant). We used the Pythagorean Theorem to calculate the standard error of the difference because estimates of $P_t$ and $H_t$ were derived from separate models. The Z-statistic, with a value of 1.96, was used because the sample size was greater than 30. Labfit software

(*Silva & Silva, 2011*) was used to determine the function that best fit the relationship between mean handling time and the frequency of observations for each forage type.

## RESULTS

Adult bulls and cows were observed 105 and 26 times, respectively (96 and 26 individuals), with a total of 64 h of feeding recorded across 32 vegetation types. Cows were observed less frequently than bulls because they tended to associate in large groups (up to 80 individuals) and were therefore more difficult to approach on foot. A total of 109 plant species were consumed.

Food types utilized were whole grass plants, grass inflorescences (only observed for cows), grass roots, whole forb plants, leaves and twigs of woody plants, bark from canopy branches of trees and shrubs, bark from the main stems of trees, bark from roots of trees and shrubs, roots of trees and shrubs, tubers (caudices), flowers and fruits. Often a trunkload was composed of more than one food type e.g. leaves and twigs or leaves and fruits.

Harvesting methods varied within and across food types. Grass plants were plucked by wrapping the trunk around the above-ground portions of a tuft and pulling to uproot the plant. If soil was attached to the roots or a significant amount of senescent leaf material was present, this was removed by thrashing the tuft against the chest or front leg. Most often the entire grass plant was consumed, but when the base of tillers was particularly robust, only the upper portion of the tuft was eaten, the roots and bases of the tillers being discarded. Grass roots were harvested in the same way except the above ground portions of the plant were discarded and only the roots eaten. Grass inflorescences were gathered by wrapping the trunk around a number of culms and pulling. Forbs with an erect growth form were plucked in a similar way to grass tufts, with the entire plant being consumed. Forbs with a creeping or climbing growth habit were gathered by extracting a long length, bundling it in the trunk, and then inserting the bundle into the mouth. Leaves of woody plants were either stripped or plucked. Stripping was most commonly done by wrapping the trunk around a leafy branch and then pulling the trunk along the length of the branch. Leaves were also stripped by loosely grasping a leafy branch in the mouth and then allowing the branch to run through the mouth while moving away from the plant. Stripping often resulted in a substantial amount of twigs being included in the trunkload. Leaves were plucked using the projections at the end of the trunk. Plucking appeared to result in fewer twigs being included in the trunkload compared to stripping, but the mass of the trunkload was potentially reduced. Leaves and twigs were harvested by wrapping the trunk around a slender branch and then bending the branch until it snapped. The entire branch was then consumed. For woody species with bark of high tensile strength (e.g. *Acacia tortilis*), leaves and twigs were harvested by grasping the end of a branch in the mouth and then drawing the branch taught across the end of a tusk until it snapped. Preference for this harvesting technique was indicated by the development of a marked groove a few centimeters back from the tip of the working tusk. Often an additional action such as breaking down a branch or felling the tree was required before a trunkload of leaves or leaves and twigs could be harvested. Bark was

**Table 1 Observations per combination of fixed effects used for modelling harvesting and handling times.**

| | Forage type | | | | | | | | |
|---|---|---|---|---|---|---|---|---|---|
| | Pluck grass | Pluck & shake mixed grass | Pluck forb | Strip leaves | Pluck leaves & twigs | Break off branch & chew off bark | Remove bark from trunk | Break off root | Dig & break off root |
| Harvesting | | | | | | | | | |
| Bull | 23 | 103 | 16 | 169 | 247 | 89 | 20 | 2 | 8 |
| Cow | 5 | 14 | – | 57 | 50 | 37 | 15 | 3 | – |
| Handling | | | | | | | | | |
| Bull | 1,524 | 99 | 1,097 | 829 | 822 | 99 | 27 | 73 | 8 |
| Cow | 150 | 10 | 168 | 126 | 201 | 32 | 18 | 7 | – |

harvested from the canopy branches of shrubs and trees by snapping off a branch (approximately 2 cm in diameter) with the trunk, placing it in the mouth and then chewing off the bark along the length of the branch. Bull elephants harvested bark from the main stems of trees by gouging and prizing out sections using their tusks. Once gouging had created a piece of bark that could be grasped by the trunk with sufficient purchase, the bark was stripped away by pulling upwards. This was only possible for tree species with bark of an adequate tensile strength. Bulls most frequently employed this technique. Cows preferred to either snap the main stem or locate a tree whose main stem had been snapped and then strip off small pieces of bark by pulling on the torn, jagged edges of bark that were created when the stem was snapped. Cows frequently employed this technique when harvesting bark from the main stems of small (main stems of approximately < 15 cm diameter) *Colophospermum mopane* trees. Roots were harvested by excavating with the feet, uprooting shrubs by plucking with the trunk, pushing over trees or by grasping exposed roots with the trunk and pulling to lift long sections out of the soil. Tubers (e.g. those of *Jatropha* spp.) were particularly sought after by cows after rain in areas with sandy soil. A unique method was used to harvest tubers. First the tuber would be partially excavated by ploughing backwards and forwards through the soil with a foot. The moist soil after rain facilitated digging because the soil did not slide back into the hole. Once part of the tuber was exposed the elephant would kneel down and impale the tuber with a tusk. The elephant would then stand up and remove the tuber from the tusk using the trunk and place it in its mouth. Fruits where either plucked from the plant or picked up from the ground after the tree had been shaken to dislodge the fruits. When gathering small fruits from the ground (e.g. pods from *Acacia tortilis*) the fruits were swept into a pile, which was then ladled into the mouth using the trunk.

Data used for the GLMM's had fewer observations for harvesting than handling (Table 1) because when elephants were feeding from a dense sward it was difficult to record precisely when harvesting began. The AIC and BIC scores indicated that the best model for harvesting included elephant ID and month as random effects, while that for handling also included vegetation type as an additional random effect (Table 2). Analysis of variance showed that both forage type and sex had a significant influence on harvesting and handling times (Table 3).

**Table 2 Results of the best GLMM's for harvesting and handling time.** The intercept, estimate (log scale) of the effect of breaking off a canopy branch and chewing off the bark, is a baseline against which the other fixed effects were compared. Forage types with positive estimates took longer to harvest or process than the baseline, and those with negative estimates took less time than the baseline. Negative estimates for sex indicate that cows had shorter harvesting and handling times than bulls.

**Harvesting time ~ Forage type + Sex + (1|Month) + (1|Elephant ID)**

| Fixed effects | Estimate | Std. Err. | t value | Pr (> |z|) |
|---|---|---|---|---|
| **Forage type** | | | | |
| Intercept (break off branch & chew off bark) | 2.906 | 0.001 | 2,345.8 | < 0.001 |
| Pluck forb | −1.153 | 0.184 | −6.3 | < 0.001 |
| Pluck grass | −1.129 | 0.001 | −910.6 | < 0.001 |
| Pluck & shake mixed grass | −0.472 | 0.001 | −369.0 | < 0.001 |
| Pluck leaves & twigs | −0.156 | 0.001 | −126.2 | < 0.001 |
| Strip leaves | −0.658 | 0.001 | −530.6 | < 0.001 |
| Remove bark from trunk | 0.638 | 0.001 | 514.7 | < 0.001 |
| Break off root | 0.221 | 0.316 | 0.7 | 0.485 |
| Dig & break off root | −0.286 | 0.001 | 955.3 | < 0.001 |
| **Sex** | | | | |
| Cow | −0.286 | 0.001 | −230.3 | < 0.001 |
| **Random effects** | **Variance** | **Std. Dev.** | | |
| Elephant ID | 0.088 | 0.296 | | |
| Month | 0.001 | 0.01 | | |

**Handling time ~ Forage type + Sex + (1|Vegetation type) + (1|Month) + (1|Elephant ID)**

| Fixed effects | Estimate | Std. Err. | t value | Pr (> |z|) |
|---|---|---|---|---|
| **Forage type** | | | | |
| Intercept (break off branch & chew off bark) | 3.889 | 0.091 | 42.85 | < 0.001 |
| Pluck forb | −1.225 | 0.066 | −18.65 | < 0.001 |
| Pluck grass | −1.204 | 0.070 | −17.18 | < 0.001 |
| Pluck & shake mixed grass | −0.644 | 0.091 | −7.06 | < 0.001 |
| Pluck leaves & twigs | −0.811 | 0.070 | −11.65 | < 0.001 |
| Strip leaves | −1.289 | 0.071 | −18.07 | < 0.001 |
| Remove bark from trunk | 0.276 | 0.116 | 2.37 | 0.018 |
| Break off root | −0.761 | 0.138 | −5.49 | < 0.001 |
| Dig & break off root | 0.035 | 0.236 | 0.15 | 0.881 |
| **Sex** | | | | |
| Cow | −0.442 | 0.098 | −4.50 | < 0.001 |
| **Random effects** | **Variance** | **Std. Dev.** | | |
| Elephant ID | 0.057 | 0.238 | | |
| Vegetation type | 0.025 | 0.16 | | |
| Month | 0.004 | 0.066 | | |

Harvesting times were short for trunkloads of green grass, forbs and leaves from woody plants (5.8–9.5 s for bulls, 4.3–7.1 s for cows); intermediate for trunkloads of mixed grass, leaves and twigs, and bark from canopy branches (11.4–18.3 s for bulls, 8.6–13.7 s for

**Table 3 Analysis of variance tables for the best GLMM's for harvesting and handling times.**

|  | Df. | Sum Sq. | Mean Sq. | F value | Chisq. | Pr (> chisq.) |
|---|---|---|---|---|---|---|
| **Harvesting time** | | | | | | |
| Forage type | 8 | 84.45 | 10.56 | 24.05 | 2,322,278.0 | < 0.001 |
| Sex | 1 | 2.78 | 2.78 | 6.33 | 53,050.0 | < 0.001 |
| **Handling time** | | | | | | |
| Forage type | 8 | 251.30 | 31.41 | 97.49 | 882.4 | < 0.001 |
| Sex | 1 | 11.96 | 11.96 | 37.12 | 20.3 | < 0.001 |

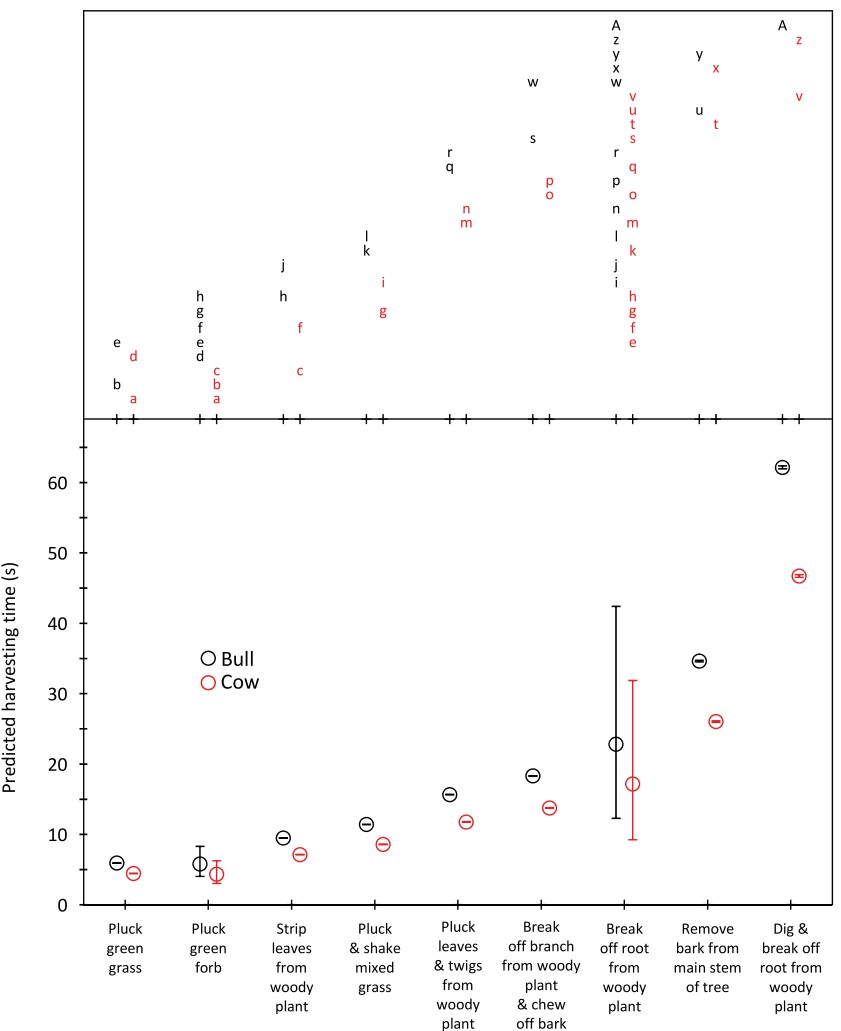

**Figure 1 Predicted harvesting times (least squares means) for adult bulls and cows across the commonly utilized forage types.** The compact letter display depicts the results of pairwise comparisons conducted using Tukey's post hoc test. Harvesting times were not significantly different (P > 0.05) for forage/gender combinations with letters in common. Bars represent 95% confidence intervals (back transformed from log scale).

cows); and long for trunkloads of roots from woody plants and main stem bark (22.8–62.1 s for bulls, 17.1–46.7 s for cows) (Fig. 1). Additional harvesting actions, such as shaking a tuft of grass to remove senescent material, significantly (*P* < 0.05) increased harvesting

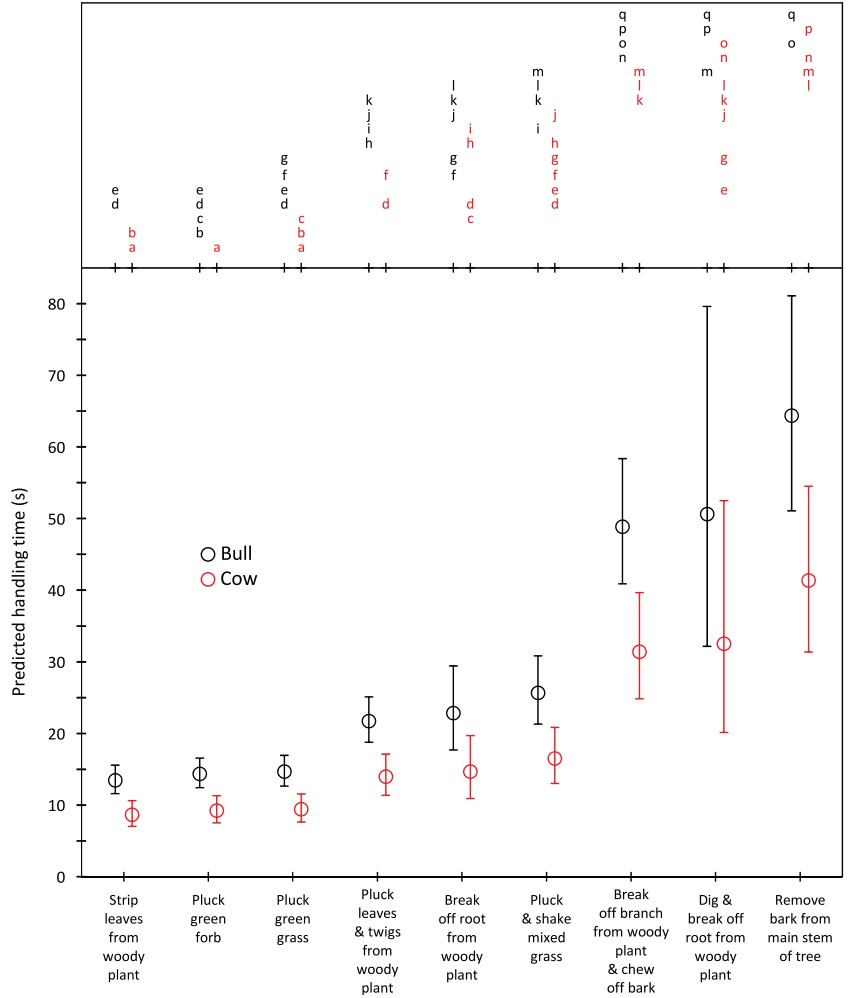

**Figure 2 Predicted handling times (least squares means) for adult bulls and cows across the commonly utilized forage types.** The compact letter display depicts the results of pairwise comparisons conducted using Tukey's post hoc test. Handling times were not significantly different ($P > 0.05$) for forage/gender combinations with letters in common. Bars represent 95% confidence intervals (back transformed from log scale).

time relative to instances when additional actions were not required. Handling times were short for trunkloads of leaves from woody plants, forbs and green grass (13.5–14.7 s for bulls, 8.7–9.4 s for cows); intermediate for leaves and twigs, roots from woody plants and mixed grass (21.7–25.7 s for bulls, 14.0–16.5 s for cows); and long for canopy bark, main stem bark and roots from woody plants that had to be excavated before being broken off (48.9–64.4 s for bulls, 31.4–41.4 s for cows) (Fig. 2). Cows had shorter harvesting and handling times ($P < 0.05$) than bulls.

Handling time was constrained by chewing for bulls, except for the processing of roots from woody plants which was limited by harvesting (Fig. 3). Time to gather a trunkload had a greater influence on handling for cows than bulls, with four out of the nine food types being constrained by harvesting as opposed only two for bulls (Fig. 4). For both

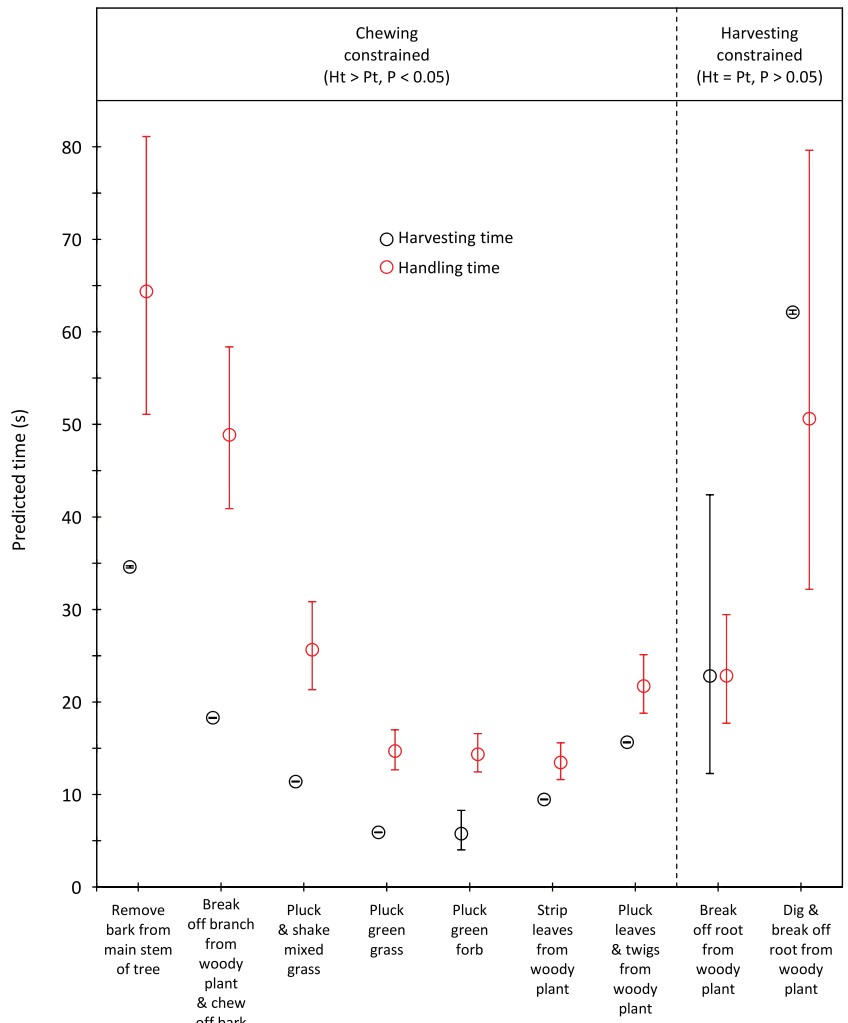

**Figure 3 Comparison of harvesting ($P_t$) and handling times ($H_t$) for adult bulls for the commonly utilized forage types.** Bars represent 95% confidence intervals (back transformed from log scale). The significance of the difference between harvesting and handling times was tested for each forage type by calculating the 95% confidence interval of the difference (intervals that included zero were not significant). Handling was assumed to be constrained by chewing when $H_t > P_t$ and by harvesting when $H_t = P_t$.

bulls and cows, trunkloads of food types with the shortest handling times were recorded most frequently (Fig. 5).

## DISCUSSION

Time to harvest and chew food has been shown to influence the intake rate of many herbivore species (for examples see *Ginnett & Demment, 1997*; *Illius et al., 2002*; *Laca, Ungar & Demment, 1994*; *Pastor et al., 1999*), but to the best of our knowledge, this is the first published study to investigate this for African savanna elephants (*Loxodonta africana*). Harvesting times for elephants were longer than those recorded for other large mammalian herbivores. For example, elephant bulls took 5.8–62.1 s to gather a trunkload, while elk (*Cervus canadensis*) and wood bison (*Bison bison athabascae*) took 0.7 and 0.5 s, respectively (*Bergman, Fryxell & Gates, 2000*; *Gross et al., 1993*). This was also the case for

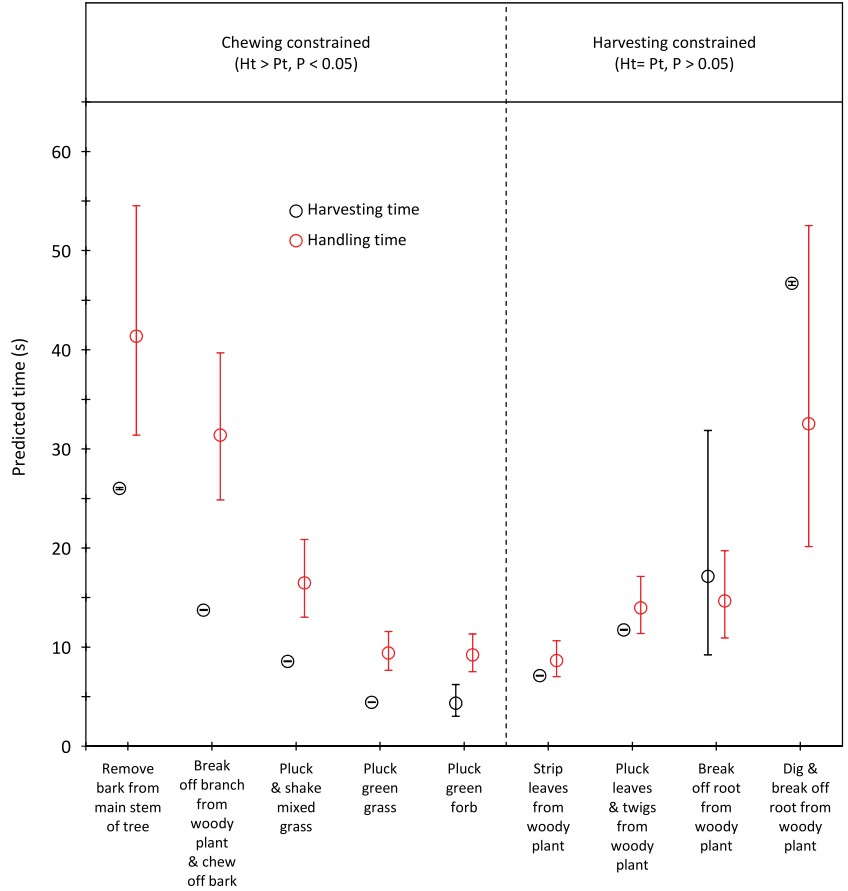

**Figure 4 Comparison of harvesting ($P_t$) and handling times ($H_t$) for adult cows for the commonly utilized forage types.** Bars represent 95% confidence intervals (back transformed from log scale). The significance of the difference between harvesting and handling times was tested for each forage type by calculating the 95% confidence interval of the difference (intervals that included zero were not significant ($P > 0.05$)). Handling was assumed to be constrained by chewing when $H_t > P_t$ and by harvesting when $H_t = P_t$.

handling time, with elephant bulls taking 13.5–64.4 s to harvest and chew a mouthful and horses (*Equus caballus*) and roe deer (*Capreolus capreolus*) taking 1.2–3 and 2.1 s, respectively (*Fleurance et al., 2009*; *Illius et al., 2002*).

Large differences in harvesting and handling times were apparent across food types. For example, bulls took three times longer to process trunkloads of main stem bark than trunkloads of leaves from woody plants. Differences in handling times are possibly more conspicuous for elephants than other herbivores because an unusually broad assortment of forage types is utilized and a particularly diverse array of harvesting methods is employed. Variation in handling time might affect the rate of intake when feeding on different food types, which may in turn influence food preferences and choice of diet (*Clegg, 2010*; *O'Connor, Goodman & Clegg, 2007*). Elephants have a fast rate of passage of ingesta (*Eltringham, 1982*). To capitalize on this, they should prefer food types that can be harvested and chewed rapidly compared to those that can only be processed more slowly (*Clegg, 2010*; *O'Connor, Goodman & Clegg, 2007*). Our observations supported this

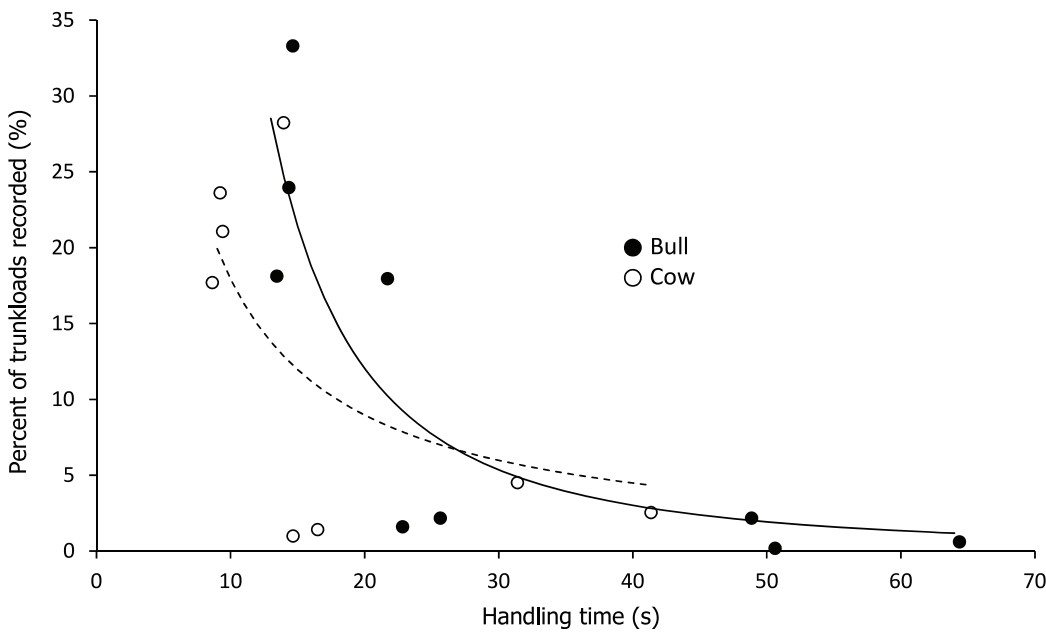

**Figure 5 Relationship between handling time and percent of total trunkloads recorded.** The relationship for bulls (●) was best represented by a second order hyperbola (solid line, $Y = a/x^2$, where $a = 2,332.246$, $P < 0.001$, adj. $R^2 = 0.729$) and for cows (○) by a first order hyperbola (dashed line, $Y = a/x$, where $a = 179.374$, $P < 0.002$, adj. $R^2 = 0.475$).

hypothesis because when all food types were available during the rainy season elephants ate predominantly green grass, forbs and leaves from woody plants (*Clegg, 2010*), which are the food types that can be harvested and chewed most rapidly. Only when these had senesced during the dry season did elephants feed more on bark and roots, which required more laborious harvesting methods and took longer to process. This seasonal change in diet has been frequently reported in the literature (*Cerling et al., 2004*; *Owen-Smith, 1988*). It is important to recognize that handling time is one of many potential constraints to food intake. Other factors such as search time (*Spalinger & Hobbs, 1992*), bite mass (*Shipely, 2007*), protein and energy content of forage (*Shrader et al., 2012*), presence of secondary metabolites and tannin-binding proteins (*Owen-Smith & Chafota, 2012*; *Schmitt, Ward & Shrader, 2016*; *Shrader et al., 2012*), and distance from a source of drinking water (*Harris et al., 2008*) also influence the relative profitability of the available food types. These are, however, considered beyond the scope of this study and will be addressed elsewhere.

Mode of harvesting had a significant effect on harvesting time. Harvesting was shorter when trunkloads could be gathered by simply plucking or stripping and longer when additional actions were necessary. For example, it took bulls almost twice as long to harvest grass tufts with a mixture of green and dry leaves compared to those with only green leaves. This was because an additional action of thrashing the plucked tuft against the chest or front leg to remove senescent material was necessary before the trunkload could be ingested. Similarly, it took twice as long to harvest a trunkload of roots from woody plants if they had to be dug up first compared to situations where they were already

exposed. This is consistent with the hypothesis that hedging of the tree layer by elephants (*Styles & Skinner, 2000*) facilitates foraging because it allows food to be harvested more rapidly and with less energy expenditure (*Smallie & O'Connor, 2000*).

Food types that could be harvested rapidly were eaten most frequently and therefore handling time was most often constrained by chewing. Under these circumstances intake rate can be increased by selecting non-fibrous plant species and parts that can be rapidly chewed. This may partially explain why elephants prefer soft, broad-leaved grasses (e.g. *Panicum maximum*), climbing forbs that don't invest heavily in structural material, and leaves with a high specific area ($cm^2 g^{-1}$) (*Clegg, 2010*; *O'Connor, Goodman & Clegg, 2007*). When rapidly harvestable food types (generally those from the herbaceous layer) are not available, handling becomes constrained by harvesting. This generally leads to increased levels of impact to woody vegetation because the additional actions required to harvest food are often destructive.

The longer handling times for bulls compared to cows were unexpected because the greater strength (body size) of bulls should allow them to harvest and chew food more rapidly. However, bulls extract larger trunkloads than cows and when this is taken into account, bulls do indeed process a greater mass of food per unit time, despite their longer handling times (*Clegg, 2010*). Harvesting methods such as pollarding or uprooting trees and using tusks to prize bark from main stems require considerable strength. Our observations suggest that these foraging techniques are largely the domain of adult bulls, presumably because their body size affords them the necessary strength. Cows appear to have fewer harvesting options available to them. This is supported by the observation that cows were often seen moving rapidly to a tree that had been felled by a bull, presumably to take advantage of a forage source that would otherwise have been inaccessible. This suggests that impact to woody vegetation should be more closely correlated to the density of adult bulls as opposed to that of the total population (*Croze, 1974*; *Guy, 1976*; *Midgley, Balfour & Kerley, 2005*). Cows compensated for their apparent lack of strength by adopting different harvesting methods to bulls. For example, they often extracted main stem bark by first snapping the trunks of small mopane trees and then stripping short lengths of inner bark from the jagged edge of the breaks. Bulls were not observed using this technique. Cows appeared to adopt a strategy of harvesting small trunkloads that allowed for rapid harvesting and chewing. This gave a sense of urgency to their feeding behavior. Bulls on the other hand appeared to be focused on larger trunkloads that took longer to harvest and chew. This difference in foraging behavior is presumably driven by the two-fold difference in body size that causes cows to have a greater energy requirement per unit body mass and bulls to have a greater absolute energy requirement per unit time (*O'Connor, Goodman & Clegg, 2007*).

## CONCLUSION

Some forage types took longer to harvest and chew than others, with both gender and the method of gathering food affecting harvesting and handling times. Handling time was mostly constrained by chewing for both sexes, but harvesting did limit processing of some food types, especially for cows. The above differences may cause variation in the rate at

which forage types can be ingested, which may in turn influence diet and habitat selection. This however can only be assessed by an intake model that also includes search time, trunkload mass, number of trunkloads harvested per patch, and the energy content of the forage as additional constraints.

## ACKNOWLEDGEMENTS

The authors thank the Malilangwe Trust for initiating the study and Julius Matsuve for assisting with data collection.

### Funding

The study was funded by the Malilangwe Trust. The funders had no role in study design, data collection and analysis, decision to publish, or preparation of the manuscript.

### Grant Disclosures

The following grant information was disclosed by the authors:
The Malilangwe Trust.

### Competing Interests

The authors declare that they have no competing interests.

### Author Contributions

- Bruce W. Clegg conceived and designed the experiments, performed the experiments, analyzed the data, contributed reagents/materials/analysis tools, wrote the paper, prepared figures and/or tables, reviewed drafts of the paper.
- Timothy G. O'Connor conceived and designed the experiments, wrote the paper, reviewed drafts of the paper.

### Animal Ethics

The following information was supplied relating to ethical approvals (i.e., approving body and any reference numbers):

Permission to conduct the study was given by the Malilangwe Trust.

### Data Deposition

The raw data has been supplied as Supplemental Dataset Files.

### Supplemental Information

Supplemental information for this article can be found online at http://dx.doi.org/10.7717/peerj.2469#supplemental-information.

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
