# Peer review of "Harvesting and chewing as constraints to forage consumption by the African savanna elephant (Loxodonta africana)"

_PeerJ, doi:10.7717/peerj.2469_

## Round 0.1 · original submission · Minor Revisions

Please address all the comments that reviewers have made promptly and explicitly. I may then need to send this back to reviewers to ensure they will be satisfied, but whether I do so will depend entirely on whether I can easily see how well you have addressed them.

·

Basic reporting

The basic reporting is good.

Experimental design

The experimental design is solid.

Validity of the findings

Due to the large sample size, the validity of the findings is not in doubt.

Additional comments

Reviewer’s Comments: Article number 11471: Clegg and O’Connor Harvesting and chewing as constraints to forage consumption by the African savanna elephant (Lox

In this paper, the authors explored whether harvesting, chewing and handing times of male and female elephants differed across food types and harvesting methods. They do this, to ultimately understanding diet choice of these megaherbivores (i.e. the aim of the study). The authors collect large samples sizes from free-ranging elephants, which allows them to explore a wide range of food types/bite types. Overall, I really enjoyed reading the ms as it provides a novel direction and insight into the foraging of elephants. As a result, it opens up a new way of looking at and the decisions that elephants use while foraging. The paper is well written, the hypotheses well thought out, the analyses appropriate, and the discussion relevant and justified. There are some areas where I think things could be expanded, and thus I have provided a few comments below that may help to improve the manuscript.

Specific Comments
Page 3 lines 52-62: Many of the references provided are quite old and thus more recent ones could be incorporated. A notable omission that could be cited throughout the ms is that of Owen-Smith, N. and J. Chafota. 2012. Selective feeding by a megaherbivore, the African elephant (Loxodonta africana). Journal of Mammalogy 93:698-705. However, there are other recent studies on elephant foraging that could also be incorporated.

Page 3 line 64: Replace Clegg 2010 (a PhD project) with some primary literature.

Page 4 line 70: Clegg and O’Connor is a paper that is in review, not in press or print. As a result, it is correct to cite it as ‘Clegg and O’Connor unpublished data’. In addition, this reference should be removed from the reference list.

Page 5 lines 95-96: This is a key point where I think the incorporation of additional data (most likely already collected in in Clegg 2010) could go a long way to telling a more complete story. Here the authors state that the aim of the study is to determine how handling time, chewing and harvesting differences between food types helps determine the diet choice of elephants. This is also presented in the abstract and in the discussion. As the diet of these elephants has most likely already been worked out by the authors, or if not then the foraging data collected in this study could easily be used to generate the elephants’ diet, a simple analysis could be done to determine the degree to which handling time, chewing and harvesting differences determine diet selection. Having done this, this would then provide the authors with a way of directly addressing the broad aim of their study.

Page 5 lines 113-115: It is unclear how this information is relevant to the study. Did it reduce food availability of certain food types for the elephants? If so, then this needs to be discussed.

Page 7 line 139: Insert ‘the’ before ‘observation’.

Page 7 line 147: Was information recorded on the species eaten with regards to forage type? If so, then these data could be used to generate the elephants’ overall diet.

Page 7 lines 150-152: What are these 9 most common combinations? Provide them here.

Page 8 line 182: What were these plant species? Can these be presented in supplementary material? I’m sure that the diet of the male and female elephants can be determined from these data. What proportion of the diet were comprised by these species? Does that diet relate to harvesting time, handling time, and/or chewing time? See points above.

Page 8 lines 183-184 & Page 9 lines 185-186: There are 11 food types listed here. How do they relate to the ‘9 most common combinations’? Explain.

Pages 9-11: Very nice descriptions of the elephants’ feeding methods.

Page 9 lines 191-194: As stated by the authors, elephants are notorious for discarding portions of the vegetation that comprises a ‘trunkful’. This is especially true for larger woody items (e.g. bark, despite the authors not reporting as an occurrence in their study). As a result, if you think about this in an optimality manner, then the benefit gained from a larger, harder to obtain item would be even less, if a portion of it is dropped and not ingested. It would be worth discussing this in the discussion section as something that would further reduce preference for the items reported to be discarded and the harder to obtain larger woody items (e.g. bark and twigs).

Page 11 lines 241-243: How are the categories of short, intermediate, and long defined? Specifically, what are the harvesting time ranges of these categories? Do these ranges overlap? Explain and define biologically. Moreover, some the categories in the text (e.g. mixed grass) do not match what is provided on the x-axis of the figure (e.g. pluck and shake grass). The same goes for the same categories for handing times (lines 246-248).

Page 12 lines 267-274: The authors make it clear that they have not obtained nutritional data, or at least it has not been included in this ms. However, there should be enough information reported in other studies to indicate that ‘green grass, forbs and leaves from woody plants’ provide greater nutritional intake (e.g. higher crude protein, lower fibre) compared to bark and roots. With that in mind, I would suggest that they indicate this here.

Page 13 lines 283-284: Define ‘hedging the tree layer’.

Page 12 line 290: What is meant by ‘a high specific area’?

Page 13 line 297-298: Bulls may process greater mass per unit time, but that assumes they eat everything that they take in their trunk, which as stated in the ms is not the case. You may find that bulls discard greater amounts of their high fibre trunkfuls, than females do of their low fibre trunkfuls and thus, the sexes may process similar amounts per unit time. Any way to comment on this?

Page 15 lines 344-345: Delete this reference as it is only a paper in review.

Figure 1: I am not familiar with compact letter displays. However, I find it hard to believe that that all the harvesting times that did not share letter combination in common differed significantly. For example, the bulls and cows in both ‘pluck forb’ and ‘break off root’ did not share letters, yet the 95% CI overlap. Should this read ‘where letters do not overlap they differ significantly’? The same is true for figure 2.
Figures 1, 3 and 4: The y-axis on Figure 1 read ‘prehension’, yet the figure caption says ‘harvesting times’. This is the first time that prehension is used in the ms (it also appears in Figures 3 and 4), and it never appear in the text. I assume that this is where the P symbol for harvesting time comes from (see page 4 lines 87-89). However, please pick one word and be consistent throughout the ms.

Figure 5: Can you please insert a legend on the figure. I see that the symbols are explained in the caption, but there are legends on all the other figures.

·

Basic reporting

This is a very good manuscript that is clearly written. There is sufficient introduction to demonstrate how this fits into a broader field of knowledge.
figures are relevant but there are a few places where the captions are inadequate.

Experimental design

The experimental design is adequate. It covers the knowledge gap well.

Validity of the findings

The data are robust and statistically sound. Conclusions are appropriate. Perhaps could have indicated how they relate to Spalinger and Hobbs' functional response paper. Some more references to appropriate literature on effects of male elephants would be appropriate, such as Guy 1976, Midgley et al. 2005.
The statement in the Abstract about weaker females is not tested and can be omitted without losing any information.

Additional comments

Very good article. Could add a little about similarities to Spalinger and Hobbs' article. Some other references would be appropriate.

---

## Round 0.2 · accepted · Accept

Thank you for producing a clear response to my reviewers' concerns. You have addressed them to my satisfaction!